# Resilience and Self-Esteem Mediated Associations between Childhood Emotional Maltreatment and Aggression in Chinese College Students

**DOI:** 10.3390/bs12100383

**Published:** 2022-10-07

**Authors:** Chen Chen, Juan Jiang, Shengkai Ji, Ying Hai

**Affiliations:** 1Center for Educational Science and Technology, Beijing Normal University, Zhuhai 519087, China; 2Department of Preschool Education, Liaoning National Normal College, Shenyang 110000, China; 3Pinghu Normal College, Jiaxing University, Jiaxing 314220, China; 4School of Educational Science, Yancheng Teachers University, Yancheng 224002, China

**Keywords:** childhood maltreatment, later development, externalizing problems, resilience, Chinese cultural context

## Abstract

Although associations between physical or sexual abuse and aggression have been mainly explored, relationships and pathways between childhood emotional maltreatment and aggression need further exploration, particularly in the Chinese cultural context. This study aimed to explore the associations between childhood emotional maltreatment and aggression and to examine the mediating effects of resilience and self-esteem on those associations. Data were obtained from a convenience sampling of 809 (aged 17–23) college students from three Chinese universities in December 2021, which was approved by the ethics committee of Beijing Normal University, China. All participants completed measures of childhood emotional maltreatment, aggression, resilience, and self-esteem. The results showed that childhood emotional maltreatment was positively associated with aggression (*r* = 0.41, *p* < 0.01), and it was negatively associated with resilience (*r* = −0.56, *p* < 0.01) and self-esteem (*r* = −0.10, *p* < 0.01). Regarding the mediation processes, resilience and self-esteem partially mediated the relationships between childhood emotional maltreatment and aggression. These findings underscore the importance of enhancing levels of resilience and self-esteem in interventions designed to reduce aggression of college students who were emotionally maltreated in childhood.

## 1. Introduction

Child maltreatment, defined as any recent act (or failure to act) which a child suffers from caregivers [1], has been confirmed to impair child development, which may contribute to it being an important worldwide public health issue [2]. Childhood emotional maltreatment (CEM), including emotional abuse and emotional neglect, reflects emotional interactions between a child and caregiver(s), and it may be the most common subtype of child maltreatment, due to its high prevalence (36%) [3].

Aggression, as a method of externalizing problems, may be an index for later maladaptation, which may be influenced by adverse childhood experiences, such as childhood maltreatment [4]. Some studies have explored relationships between physical or sexual abuse and aggression [5,6] and relationships between CEM and aggression have also been confirmed by some studies [7], while the pathways from CEM to aggression remain unclear, particularly in the Chinese cultural context. Attachment theory posits that early interactions between a child and caregiver(s) may influence individuals’ internal working models about themselves, others, and interpersonal relationships, which may influence later development [8]. Guided by attachment theory, therefore, the current study attempted to explore relationships between CEM and aggression and to examine the pathways from CEM to aggression in the Chinese cultural context.

### 1.1. Childhood Emotional Maltreatment and Aggression

CEM refers to the hostile rejecting of the child, degrading, corrupting, or denying emotional responsiveness, or failing to provide for the child’s emotional needs [9], which may contribute to later maladaptation, such as aggression. For example, a study reported that CEM was positively associated with aggression in a sample of Chinese college students [10]. Another study confirmed these relationships based on 1266 Chinese college students [11]. Moreover, some studies found these relationships in samples of Chinese adolescents [7,12]. Similarly, these relationships between CEM and aggression have also been confirmed in Western adult samples [13,14]. Additionally, A study reported that relationships between childhood maltreatment and aggression were not influenced by types of childhood maltreatment based on a meta-analysis [15], while other study found that emotional maltreatment was more related to internalizing than externalizing behaviors (e.g., aggression) in Chinese samples based on a meta-analysis [16]. A study reported that CEM did not predict individuals’ aggression based on a longitudinal study [17].

Although several studies have explored relationships between CEM and aggression, this topic still needs further exploration. Individuals who have suffered CEM may be inefficient in emotional regulation and raise their selective attention to threat-related signals (i.e., anger-related signals), which may contribute to low levels of control of feelings such as anger [9,18], and these emotional dysregulations and disorganized anger may lead to high levels of aggression [19]. Meanwhile, a study suggested that individuals who had experienced CEM had some negative schemas, such as mistrust, self-sacrifice, and emotional inhibition, which may lead to high levels of aggression [20]. The college period is one of the most important periods in a person’s life, and experiences in childhood may have more effects on individuals’ development in this period. Similarly, being a college student may be an important component for social development, particularly in developing countries, which suggests that we should pay attention to college students’ individual development, such as aggression. The current study, therefore, attempted to explore the relationships between CEM and aggression in the Chinese cultural context. We hypothesized that CEM would be found to have a positive relationship with aggression in a sample of Chinese college students.

### 1.2. Resilience and Self-Esteem as Mediators

Resilience is an ability that helps people master normative developmental tasks despite significant negative experiences [21], and this ability may be influenced by adverse childhood experiences [22,23]. For example, a study reported that Chinese college students who have experienced childhood maltreatment might have low levels of resilience [24]. A study found that emotional maltreatment was negatively associated with resilience in a sample of Chinese college students [25]. Similarly, relationships between childhood maltreatment and aggression have also been confirmed in Western samples based on longitudinal designs [26]. Moreover, relationships between resilience and aggression have been explored by some studies. For instance, a study found that resilience was negatively associated with aggression in a sample of Chinese college students [23]. Further, a study confirmed these relationships in a sample of Western adolescents [27]. Additionally, a meta-analysis also confirmed the relationships between adverse childhood experiences and resilience [28].

Meanwhile, self-esteem (SE), including positive or negative evaluations about oneself, is an important construct in individuals’ development [29], and it may be influenced by adverse childhood experiences. Some studies have explored relationships between child maltreatment and self-esteem. For example, a study reported that childhood maltreatment was negatively associated with self-esteem in 324 Chinese college students [30]. A study found that emotional maltreatment was negatively associated with self-esteem in a sample of Chinese college students [31]. These relationships were also confirmed in Chinese adolescents and Western young adults [32,33]. Similarly, some studies suggested that low levels of self-esteem may be associated with aggression [34,35].

Although relationships between child maltreatment, aggression, resilience, and self-esteem have been explored by some studies, roles of resilience and self-esteem in the relationships between child maltreatment and aggression need further exploration, particularly in the relationships between CEM and aggression. Moreover, resilience and self-esteem have been found to mediate the relationships between child maltreatment and later development, respectively [6,36], which suggests that resilience and self-esteem may be potential mediators in adverse childhood experiences and later maladaptation [37,38]. Therefore, the current study attempted to examine roles of resilience and self-esteem in the relationships between CEM and aggression. Additionally, CEM, a kind of adverse childhood experiences, influences the development of self and relationships with others (e.g., low levels of self-esteem and resilience; [28,30]), and this biased self and these relationships with others may be bridges for later maladaptation. Therefore, the current study attempted to explore roles of resilience and self-esteem in the relationships between CEM and aggression. We hypothesized that resilience and self-esteem would be found to parallel mediate relationships between CEM and aggression in a sample of Chinese college students.

## 2. The Current Study

Although relationships between CEM and aggression have been explored by some studies, the pathways from CEM to aggression still need further exploration. Among countries, China has the largest population in the world, and Chinese behaviors may be influenced by the traditional Chinese culture, which emphasizes hierarchy in families, which may raise the importance of exploring CEM. What relationships between CEM and aggression are and how CEM influences aggression may be two important issues for child maltreatment studies in the Chinese cultural context. The current study, therefore, attempted to (a) explore relationships between CEM and aggression, and (b) examine roles of resilience and self-esteem in relationships between CEM and aggression in a sample of Chinese college students. We hypothesized that CEM would be found to have a positive relationship with aggression, and resilience and self-esteem would parallel mediate the relationships between CEM and aggression in a sample of Chinese college students. Specifically, individuals who have experienced CEM may have high levels of aggression, and high levels of resilience and self-esteem may decrease the effects of CEM on aggression in a sample of Chinese college students.

## 3. Method

### 3.1. Participants

A total of 809 college students were recruited from three universities in Liaoning Province of China with convenience sampling techniques in December, 2021. The mean age of participants was 19.07 years (SD = 1.17), and 76.5% (619/809) of them were women. A total of 61.20% (495/809) of them were the only child in their family. The detailed demographic information is presented in Table 1.

### 3.2. Measures 

The Childhood Trauma Questionnaire—Short Form (Appendix A) (CTQ-SF; [39]).

The CTQ-SF, a 28-item self-reported screening tool, has been used to assess childhood maltreatment widely. It has five subscales, namely physical abuse (e.g., *Punished with a belt*), emotional abuse (e.g., *Called names*), sexual abuse (e.g., *Touched in a sexual way*), physical neglect (e.g., *Wear dirty clothes*), and emotional neglect (e.g., *Loved each other*). Each item is rated via a five-point Likert scale (from 1 = *never* to 5 = *always*), and high scores indicate high levels of childhood maltreatment. The Chinese adaptation of the CTQ-SF was developed by Zhao and colleague [40], and has been used widely in the Chinese cultural context [41]. The subscales of emotional abuse (EA) and emotional neglect (EN) were administered in the current study to assess CEM in Chinese college students, and Cronbach’s alpha was 0.75 and 0.84, respectively.

The Aggression Questionnaire (Appendix A) (AQ; [42]).

The AQ, a 29-item self-reported scale that has been used to assess aggression, has been adapted into Chinese by Li and colleague [43] with five subscales, namely physical aggression (e.g., *I have threatened people I know*), verbal aggression (e.g., *My friends say that I am somewhat argumentative*), anger (e.g., *Some of my friends think I am a hothead*), hostility (e.g., *I am sometimes eaten up with jealousy*), and self-aggression (e.g., *I have harmed myself*). Each item is responded via a five-point Likert scale (from 1 = *not at all characteristic* to 5 = *very characteristic*), and high scores indicate high levels of aggression. The Chinese version of the AQ has been widely used in Chinese samples [44]. The subscales of physical aggression (PA), verbal aggression (VA), anger (AN), and hostility (HO) were used in the current study to assess the levels of aggression in Chinese college students, and Cronbach’s alpha was 0.72, 0.83, 0.81, and 0.71, respectively.

Rosenberg Self-Esteem Scale (Appendix A) (SES; [45]).

The SES, a 10-item self-reported questionnaire that has been used to assess individuals’ self-esteem, has been widely used in the Chinese cultural context [46]. Participants rate each item on a four-point Likert scale (from 1 = *very low conformity* to 4 = *very high conformity*), and high scores indicate high levels of self-esteem (SE). The Chinese version of SES was administered in the current study to assess the levels of self-esteem in Chinese college students, and Cronbach’s alpha was 0.92.

The Chinese Adolescents Resilience Scale (Appendix A) (CARS; [47]).

The CARS, a 27-item self-reported scale that has been used to assess individuals’ resilience, contains five subscales, namely the goal-focused (e.g., *My life has clear purposes*), emotional control (e.g., *I am always discouraged by failures*), positive cognition (e.g., *I think adversity motivate me*), family support (e.g., *My parents respect my opinions*), and interpersonal assistance subscales (e.g., *I will ask for help when facing difficulties*). Each item is rated via a five-point Likert scale (from 1 = *very low conformity* to 5 = *very high conformity*), and high scores indicate high levels of resilience. The CARS was administered in the current study to assess levels of resilience in Chinese college students, and Cronbach’s alpha was 0.85.

### 3.3. Procedure

There were several steps for collecting data in the current study. First, the first author presented the aims and processes of the current study to four potential cooperators, and three of them agreed to collect data in their universities. Second, the authors, respectively, went to these three universities and sent questionnaires to 900 participants initially. All participants were informed about the research before data collection process, and signed informed consent attached at the beginning of the measurement booklets, which informed them about the purposes of the study and assured them that their answers would only be used anonymously for research purposes. Third, all participants completed questionnaires in classrooms within 20 min, and all of them received a small gift worth 5 RMB (USD 0.74). Fourth, the research assistants and the authors completed the data cleaning and analysis.

### 3.4. Ethical Approval

The study was approved by the ethics committee of Faculty of Education, Beijing Normal University, China, and all materials and procedures were safe for participants.

### 3.5. Data Analysis

First, missing values and outliers were examined before data analysis, and questionnaires with 15% missing data were removed in data analysis [48]; a total of 809 participants were included in data analysis process. Second, a Pearson correlation analysis was used to examine relationships between all variables. Third, the direct effect model of the structural equation modeling (SEM) was performed to examine relationships between CEM and aggression. Fourth, the mediating effects models of resilience and self-esteem were examined by SEM, and Bootstrapping method with 1000 times repeated-sampling was used to examine mediating effects. All analyses were conducted by IBM SPSS Statistics for Windows, Version 22.0 [49] and IBM SPSS AMOS for Windows, Version 22.0 [50], and all tests were two-tailed for significance with *p* value set at 0.05. Moreover, multiple model fit indexes were used to compare models in the current study, including root-mean-square error of approximation (RMSEA, 0.080 or less), standardized root-mean-square-residual (SRMR, 0.080 or less), comparative fit index (CFI, 0.900 or more), normative fit index (NFI, 0.900, or more), goodness fit index (GFI, 0.900 or more), and Tucker–Lewis index (TLI, 0.900 or more) [51].

## 4. Results

### 4.1. Descriptive Analysis

Descriptive statistics and correlations are presented in Table 2. As indicated, CEM was positively associated with aggression (*r* = 0.41, *p* < 0.01), and was negatively associated with resilience (*r* = −0.56, *p* < 0.01) and self-esteem (*r* = −0.10, *p* < 0.01). Aggression was negativity associated with resilience (*r* = −0.44, *p* < 0.01) and self-esteem (*r* = −0.17, *p* < 0.01).

### 4.2. Effects of Childhood Emotional Maltreatment on Aggression

Average scores were used to determine CEM, self-esteem, resilience and aggression constructs, and all coefficients were standardized estimates. SEM was used to measure the direct relationship between CEM and aggression. The results of the SEM indicated that CEM was positively associated with aggression (*β* = 0.43, SE = 0.031, *p* < 0.001). Further, the model showed an acceptable fit to the data [52] (see Table 3).

### 4.3. Mediation Analysis

SEM was used to measure the indirect relationships among CEM, resilience, self-esteem and aggression. Further, the Bootstrapping method was used to examine the mediating effects. The SEM results showed that the structural model provided a good fit to the data (see Table 3). The SEM results indicating the relationships between variables are presented in Figure 1 and Figure 2.

In this mediation model (Figure 2), CEM was negatively associated with resilience (*β* = −0.62, *p* < 0.001) and self-esteem (*β* = −0.11, *p* < 0.01), and was positively associated with aggression (*β* = 0.24, *p* < 0.001). Resilience was negatively associated with aggression (*β* = −0.28, *p* < 0.001). Furthermore, self-esteem was negatively associated with aggression (*β* = −0.12, *p* < 0.001). Both of the 95% confidence intervals of direct and indirect effects did not include 0 (see Table 3). Thus, resilience and self-esteem partially mediated the relationships between CEM and aggression in Chinese college students.

## 5. Discussion

The current study explored relationships between CEM and aggression and examined roles of resilience and self-esteem in those relationships in the Chinese cultural context, which may broaden the scope of child maltreatment studies within multiple cultures and provide possible explanations of pathways from CEM to aggression. The findings highlight the importance of increasing levels of resilience and self-esteem of survivors of CEM in the Chinese cultural context.

The results showed that CEM was significantly positively associated with aggression, findings which were consistent with previous studies [11,15,16]. Individuals who have suffered CEM may have difficulty in emotional regulation for improper or deficit emotional interactions with caregivers [53], and they may have a hyperactive amygdale, which may contribute to high levels of aggression [54]. Moreover, they may have insecure attachment with their caregivers, which may establish biased internal working models about interpersonal relationships [55,56]. Additionally, while Chinese people may be different from Western people, the findings in the present study may suggest that relationships between CEM and aggression exist across nations and cultures.

Moreover, the results showed that CEM was negatively associated with resilience and self-esteem, findings which were consistent with previous studies [24,28,31]. Individuals who have experienced CEM may feel more loneliness [57] and they may have less skills to communicate with others [58], which may contribute to their low levels of resilience. Meanwhile, individuals who have experienced CEM may be estranged to others or have social anxiety about their communication skills [59], and they may not have the ability to judge whether the behaviors towards or evaluations about them from their caregivers are right or not, which may contribute to their improper evaluations or ideas of themselves [60]. Similarly, individuals that suffered CEM may have disorganized attachment with their caregivers, which may contribute to their biased evaluations of self, such as low levels of self-esteem [61].

Furthermore, the results demonstrated that resilience and self-esteem parallel mediated the relationships between CEM and aggression, suggesting that CEM not only has direct effects on aggression, but also has indirect effects though resilience and self-esteem. These results also suggested that resilience and self-esteem may play roles in the relationships between child maltreatment and later development, which has also been suggested by the results previous studies [6,38]. Moreover, Chinese people prefer constructing a self based on interpersonal relationships [62], which may contribute to severe impacts of CEM on resilience and self-esteem. Survivors of CEM with high levels of resilience and self-esteem may overcome the influence of CEM, which may contribute to low levels of aggression. These findings suggest that strategies for increasing resilience and self-esteem should be conducted to help survivors of CEM overcome the impact of CEM on later development in the Chinese cultural context.

### 5.1. Limitations

Several limitations should be acknowledged in this study. First, a retrospective study design and self-reported scales were used, which may have influenced the accuracy of information. Future research should take participants’ age and memory biases into consideration, and collect data based on several resources (e.g., peers, parents, and teachers). Second, the current study used convenience sampling techniques to recruit the participants, which may not have been a representative sample. Future studies should recruit participants based on random sampling or stratified sample techniques. Third, a cross-sectional design was used to examine the pathways from CEM to aggression, which may not have provided solid evidence for mechanisms linking CEM to aggression. Longitudinal studies should be conducted in the future to confirm the results of the current study. Fourth, the participants were recruited from China, and Chinese people may be quite different to Western cultures, which may influence the ability of the results to be generalized to all cultural contexts. Future studies should recruit participants from different cultural contexts and examine the relationships between CEM and aggression across nations and cultures. Fifth, individuals who have experienced CEM may have poor academic achievement, which may contribute to their low levels of education. The current study explored relationships between CEM, aggression, self-esteem, and resilience in a sample of Chinese college students, which may not provide the whole picture of those relationships. Future studies should recruit non-university adolescents or young adults, which may help us better understand the pathways from CEM to aggression.

### 5.2. Strengths

Although the current study had several limitations, it also had several strengths. First, the study verified relationships between CEM and aggression in a sample of Chinese college students, which may broaden the scope of childhood maltreatment studies across nations and cultures. Second, the study examined roles of resilience and self-esteem in the relationships between CEM and aggression, which may provide possible mechanisms linking childhood maltreatment to later development. Third, the study may provide possible strategies for CEM interventions in college students.

### 5.3. Practical Implications

Keeping the limitations and strengths of the current study in mind, its implications should be also acknowledged. First, the results showed that CEM was positively associated with aggression in college students, which suggests that CEM may have a long-term effect on individuals’ later development. Governments and communities should provide support for positive parenting behaviors of Chinese parents and decrease the prevalence of CEM in the Chinese population. Second, the results showed that resilience and self-esteem parallel mediated the relationships between CEM and aggression, which suggests that resilience and self-esteem may be possible mediators of pathways from CEM to aggression. Schools should provide strategies for increasing levels of resilience and self-esteem in survivors of CEM. For example, schools should provide courses about communication skills, which may increase levels of communication skills in survivors of CEM, and in turn, contribute to high levels of resilience. Additionally, teachers may give more positive feedback to survivors of CEM and help them rebuild their evaluations of self, which may increase levels of self-esteem in survivors of CEM.

## 6. Conclusions

The current study explored relationships between CEM and aggression and examined the roles of resilience and self-esteem in those relationships in a sample of Chinese college students, which may broaden the scope of childhood maltreatment studies across nations and cultures and may provide possible mechanisms linking childhood maltreatment to later development in the Chinese cultural context. The results showed that childhood emotional maltreatment was positively associated with aggression and negatively associated with resilience and self-esteem. Moreover, resilience and self-esteem parallel mediated the relationships between childhood emotional maltreatment and aggression, a finding which was in line with the hypothesis. The findings underscore the importance of enhancing resilience and self-esteem in interventions designed to reduce the aggression of college students who were emotionally maltreated in childhood in the Chinese cultural context.

## Figures and Tables

**Figure 1 behavsci-12-00383-f001:**
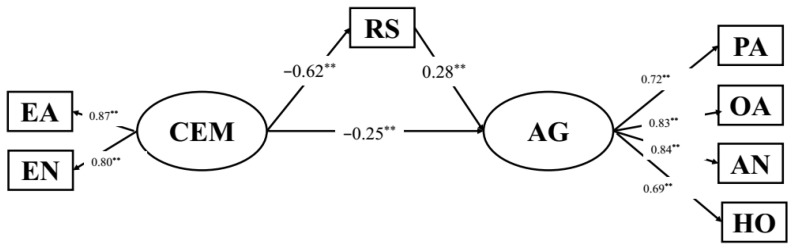
Standardized parameter estimates of the structural model demonstrating effects of CEM on aggression via resilience. Note: N = 809; EA: emotional abuse; EN: emotional neglect; PA: physical aggression; VA: verbal aggression; AN: anger; HO: hostility; CEM: childhood emotional maltreatment; AG: aggression; RS: resilience, ** *p* < 0.01.

**Figure 2 behavsci-12-00383-f002:**
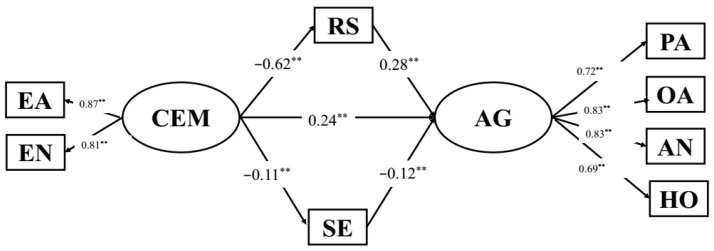
Standardized parameter estimates of the structural model demonstrating effects of CEM on aggression via resilience and self-esteem. Note: N = 809; EA: emotional abuse; EN: emotional neglect; PA: physical aggression; VA: verbal aggression; AN: anger; HO: hostility; CEM: childhood emotional maltreatment; AG: aggression; RS: resilience; SE: self-esteem, ** *p* < 0.01.

**Table 1 behavsci-12-00383-t001:** Basic information of participants (*n* = 809).

Variables	Gender		Child Number		Hometown			Family Month—Income (RMB)			
	Male	Female	One	>One	Countryside	Town	City	Below 1000	1000–3000	3001–5000	Above 5000
%	23.49	76.51	61.19	38.81	28.92	25.83	45.25	8.53	37.33	23.98	30.16

**Table 2 behavsci-12-00383-t002:** Means, standard deviations, and correlations between all variables (*n* = 809).

Variables		Male		Female		*t*	*p*	95% CI		1	2	3	4	5
		M	SD	M	SD			Lower Bound	Upper Bound					
1	Age	19.09	1.08	19.06	1.20	2.88	0.785	−0.164	0.217	-				
2	CEM	2.17	0.77	1.76	0.75	6.43	0.000	0.278	0.522	0.03	-			
3	AG	2.42	0.56	2.32	0.58	2.11	0.035	0.005	0.194	0.07	0.41 **	-		
4	RS	3.39	0.49	3.53	0.51	−3.40	0.001	−0.226	−0.061	−0.10 **	−0.56 **	−0.44 **	-	
5	SE	2.63	0.71	2.61	0.80	0.40	0.692	−0.102	0.153	0.05	−0.10 **	−0.17 **	0.10 **	-

Note: CEM: childhood emotional maltreatment; AG: aggression; RS: resilience; SE: self-esteem. ** *p* < 0.01.

**Table 3 behavsci-12-00383-t003:** Mediation effects of childhood emotional maltreatment on aggression via resilience and self-esteem (*n* = 809).

Models	Model Fit Indexes								Direct/Indirect Effects	Bootstrap Confidence	
	*χ*^2^/*df*	RMSEA	SRMR	CFI	NFI	GFI	TLI	RFI		Lower Bounds	Upper Bounds
CEM-AG	6.677	0.084	0.019	0.978	0.975	0.979	0.960	0.953	-	-	-
CEM-RS-AG	5.279	0.073	0.017	0.981	0.977	0.980	0.964	0.956	Indirect Effect	0.081	0.170
									Direct Effect	0.096	0.260
CEM-RS-SE-AG	4.578	0.056	0.017	0.977	0.971	0.968	0.960	0.949	Indirect Effect	0.089	0.178
									Direct Effect	0.088	0.246

Note: CEM: childhood emotional maltreatment; AG: aggression; RS: resilience; SE: self-esteem. Bootstrap Confidence: Bias-corrected percentile method, 95%CI.

## Data Availability

Not applicable.

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
