# Peer review of "Resilience and Self-Esteem Mediated Associations between Childhood Emotional Maltreatment and Aggression in Chinese College Students"

_behavsci, 2022, doi:10.3390/bs12100383_

Round 1

Reviewer 1 Report

Based on a college student sample this paper analyzes associations between childhood emotional maltreatment and aggression and examines the mediating roles of resilience and self-esteem in the associations between childhood emotional maltreatment and aggression.

---Specific comments---

1. I consider it important that the authors specify the study population, that is, college students, both in the title of the manuscript and in the keywords.

2. In general terms, I noted that the literature review included research carried out in childhood, adolescence and youth population. However, the study sample is base on exclusively college students’ population (M = 19.07). So, I recommend authors review this section and focused on the population under study and more specifically, in youth population of Asian context. If there is not enough literature, then please organize the information by developmental stages, variables and contexts.

3. In current study section, please, make sure all objectives and hypotheses are stated in this section and explain the results expected.

4. Please describe the procedure in more detail. For example, I wonder, how was the data collection process? How long did the data collection take? Were the questionnaires on paper or online? were they collected inside the classrooms? the authors should be clearer in this respect.

5. In the Discussion section, again, I do not read clearly if the studies cited by the authors and that are in line with the findings of the manuscript have focused on young people, adolescents or children. Besides, it is known that Asian population is quite different to other contexts such as American context or European context. So, I suggest review the studies cited and add a cultural lens in this section. In general, I recommend the authors to review this section and make the appropriate modifications being clearer regarding the population of the studies cited and adding a cultural lens as a possible explanation for the findings.

6. Please, revise the limitations of the study. The participants of the study are Asian population. Thus, this aspect should be a limitation of the study in order to avoid to generalize the results to all contexts population.

7. Moreover, it is important that the authors highlight the strengths of the study and not only limitations. At the end of the manuscript, conclusions should be explained in details. I wonder, in what ways does the manuscript contribute to the field of study?

8. Please, revise References section according to ACS style. Sometimes Doi is missing. For instance:

Buss, A. H., & Perry, M. (1992). The aggression questionnaire. Journal of Personality and Social Psychology, 63(3), 334 452–459.

Author Response

Dear Editors and Reviewers,

Thank you for your letter and for the reviewer’s comments concerning our manuscript entitled “Resilience and self-esteem mediated associations between childhood emotional maltreatment and aggression” (Manuscript ID: behavsci-1858696) in Behavioral Sciences. Those comments are all valuable and very helpful for revising and improving our manuscript, as well as the important guiding significance to our future studies. We have studied the comments carefully and have made corrections which we hope meet with approval.

The main corrections in the manuscript and the responses to the reviewers’ comment are as follows.

Responses to the reviewers’ comments:

Reviewer#1: Comments and Suggestions for Authors

Comment 0: Based on a college student sample this paper analyzes associations between childhood emotional maltreatment and aggression and examines the mediating roles of resilience and self-esteem in the associations between childhood emotional maltreatment and aggression.

---Specific comments---

Comment 1: I consider it important that the authors specify the study population, that is, college students, both in the title of the manuscript and in the keywords.

Response: Thank you for your comments and suggestions. We have revised these issues in the manuscript, please find the detailed information in the revised manuscript.

Comment 2: In general terms, I noted that the literature review included research carried out in childhood, adolescence and youth population. However, the study sample is based on exclusively college students’ population (M= 19.07). So, I recommend authors review this section and focused on the population under study and more specifically, in youth population of Asian context. If there is not enough literature, then please organize the information by developmental stages, variables and contexts.

Response: Thank you for your comments and suggestions. We have added more studies which based on youth population of Asian context to the manuscript, please find the detailed information in the revised manuscript.

Comment 3: In current study section, please, make sure all objectives and hypotheses are stated in this section and explain the results expected.

Response: Thank you for your comments and suggestions. We have revised the section of in current study, please find the detailed information in the revised manuscript.

Comment 4: Please describe the procedure in more detail. For example, I wonder, how was the data collection process? How long did the data collection take? Were the questionnaires on paper or online? were they collected inside the classrooms? the authors should be clearer in this respect.

Response: Thank you for your comments and suggestions. We have revised these issues in the manuscript and presented a separate section of procedure in the method part, such as “There were several steps for collecting data in the current studies. First, the first author presented the aims and processes of the current study to four potential cooperators, and three of them agreed to collect data in their universities. Second, the authors, respectively, went to these three universities and sent questionnaires to 900 participants initially. All participants were acknowledged about the research before data collection process, and informed consent attached at the beginning of the measure booklets, which to inform them about the purposes of the study and ensure them that their answers would only be used anonymously for research purposes. Third, all participants completed the questionnaires in classrooms within 20 minutes, and all of them received a small gift worth 5RMB ($0.74).”.

Comment 5: In the Discussion section, again, I do not read clearly if the studies cited by the authors and that are in line with the findings of the manuscript have focused on young people, adolescents or children. Besides, it is known that Asian population is quite different to other contexts such as American context or European context. So, I suggest review the studies cited and add a cultural lens in this section. In general, I recommend the authors to review this section and make the appropriate modifications being clearer regarding the population of the studies cited and adding a cultural lens as a possible explanation for the findings.

Response: Thank you for your comments and suggestions. We have cited more studies based on young people in the Discussion part. Moreover, we have added a cultural lens to explain the findings.

Comment 6: Please, revise the limitations of the study. The participants of the study are Asian population. Thus, this aspect should be a limitation of the study in order to avoid to generalize the results to all contexts population.

Response: Thank you for your comments and suggestions. We have added this limitation to  the manuscript, please find the detailed information in the revised manuscript.

Comment 7: Moreover, it is important that the authors highlight the strengths of the study and not only limitations. At the end of the manuscript, conclusions should be explained in details. I wonder, in what ways does the manuscript contribute to the field of study?

Response: Thank you for your comments and suggestions. We have revised these issues in the manuscript, please find the detailed information in the revised manuscript.

Comment 8: Please, revise References section according to ACS style. Sometimes Doi is missing. For instance:

Buss, A. H., & Perry, M. (1992). The aggression questionnaire. Journal of Personality and Social Psychology, 63(3), 334 452–459.

Response: Thank you for your comments and suggestions. We have checked the references and revised these issues in the manuscript, please find the detailed information in the revised manuscript.

Reviewer 2 Report

The present study is relevant and well-designed, but some major improvements/updates are necessary. 

-          Please proofread the paper.

-          I recommend a proofreading by an English speaker.

Abstract

- “Data were obtained from a 18 convenience sampling of (aged 17-23) college students.” When? Where?

- Ethical approval?

- “The results showed that childhood emotional maltreatment positively associated with aggression, and negatively associated with resilience and self-esteem.” Please present some statistical findings according to APA guidelines.

Keywords: please use the maximum number of keywords according to instructions to authors; please use some MeSH terms; Preferably, keywords should be different from those used in abstract or title.

-          “1. ,Introduction” or “1. Introduction”

-          (36%; Stoltenborgh et 39 al., 2012) or (36%) (Stoltenborgh et 39 al., 2012).?

-          “Aggression, as a behavior problem, may be an index for maladaptation, which may be influenced by adverse experiences.” Please give more details. This sentence is not clear.

-          “CEM and aggression are still unclear”; Why? Please give more details. Have you carried out a review? (more than 206000 studies); please see https://scholar.google.com/scholar?hl=pt-PT&as_sdt=0%2C5&q=Childhood+emotional+maltreatment&btnG=

-          Please present study objectives, research questions and study hypothesis all together at the end of introduction.

-          Lines 99-102: please avoid this type of sentences since authors have not carried out a systematic review. Please rephrase: “Although the relationships between child maltreatment, aggression, resilience, and self-esteem have been explored by some studies, the roles of resilience and self-esteem in the relationships between child maltreatment and aggression are unclear, particularly CEM.” Unclear? Why? References? Please not use this type of sentences. References are missing.

-          Please not use this sentence: “few studies have explored the parallel mediating effects of resilience and self-esteem int he relationships between CEM and aggression” Few studies? How do you know? Please not use this type of sentences without developing a strong argumentation. References are missing.

-          “Although the relationships between CEM and aggression has been explored by some studies, the pathways from CEM and aggression are still unclear.” Please rewrite. References are also missing.

-          Please cite some systematic reviews on the present topic (or related topics) in introduction. The new references should also be cited in discussion.

Methods

-          Please create a separated section for Ethical approval

-          3.1. Participants and procedure

-          “The current study recruited 900 participants from three universities in Liaoning Province of China”; When?

-          Please present the administered tools in a supplementary file, such as Childhood Trauma Questionnaire–Short Form, the CTQ-SF or the Aggression Questionnaire, etc.

-          Methods are incomplete, please present how each one of these tools were administered. Self-administration? Where? When? How? Who?

-          Please consider the presentation of point 3.2. Measures in a Tabular format.

-          3.3. Data analysis: SPSS 22 and AMOS 22 – please correctly cite SPSS and AMOS. For instance see, https://www.ibm.com/support/pages/how-cite-ibm-spss-statistics-or-earlier-versions-spss

-          How were the models validated?

4. Results

- Please check the presentation of all statistical findings in APA guidelines.

- Please check the format of figures and Tables in instructions to authors.

5. Discussion

-          Please cite some new systematic reviews on the present topic

-          Please cite more related or similar studies in discussion

-          Please clearly reply to study objectives, research questions and study hypothesis in discussion.

-          Please create a section about study strengths at the end of discussion; please explain in this section. What this study adds in relation to the state of art?

Conclusion

-          Conclusion should reply to study objectives.

-          “Findings underscore the  importance of enhancing resilience and self-esteem in interventions  designed to reduce aggression of college students who were emotion-ally maltreated in childhood.” Please create a section about practical implications at the end of discussion.

References

-          Please check the format of references in instructions to authors.

Author Response

Dear Editors and Reviewers,

Thank you for your letter and for the reviewer’s comments concerning our manuscript entitled “Resilience and self-esteem mediated associations between childhood emotional maltreatment and aggression” (Manuscript ID: behavsci-1858696) in Behavioral Sciences. Those comments are all valuable and very helpful for revising and improving our manuscript, as well as the important guiding significance to our future studies. We have studied the comments carefully and have made corrections which we hope meet with approval.

The main corrections in the manuscript and the responses to the reviewers’ comment are as follows.

Responses to the reviewers’ comments:

Reviewer#2: Comments and Suggestions for Authors

Comment 1: The present study is relevant and well-designed, but some major improvements/updates are necessary. Please proofread the paper. I recommend a proofreading by an English speaker.

Response: Thank you for your comments and suggestions. We have asked an English speaker to check the language throughout the manuscript, please find the detailed information in the revised manuscript.

Abstract

Comment 2: - “Data were obtained from a convenience sampling of (aged 17-23) college students.” When? Where?

Response: Thank you for your comments and suggestions. We have added this information to the revised manuscript.

Comment 3: - Ethical approval?

Response: Thank you for your comments and suggestions. We have added this information to the revised manuscript.

Comment 4: - “The results showed that childhood emotional maltreatment positively associated with aggression, and negatively associated with resilience and self-esteem.” Please present some statistical findings according to APA guidelines.

Response: Thank you for your comments and suggestions. We have added this information to the revised manuscript.

Comment 5: Keywords: please use the maximum number of keywords according to instructions to authors; please use some MeSH terms; Preferably, keywords should be different from those used in abstract or title.

Response: Thank you for your comments and suggestions. We have revised and added the keywords to the revised manuscript, such as “childhood maltreatment; later development; externalizing problems; maladaptation; resilience; social support; self-evaluation; self; mediation roles; Chinese cultural context”.

Comment 6: -“1. ,Introduction” or “1. Introduction”

Response: Thank you for your comments and suggestions. We have revised this issue in the revised manuscript.

Comment 7: -(36%; Stoltenborgh et 39 al., 2012) or (36%) (Stoltenborgh et 39 al., 2012).?

Response: Thank you for your comments and suggestions. We have revised this issue in the revised manuscript.

Comment 8: - “Aggression, as a behavior problem, may be an index for maladaptation, which may be influenced by adverse experiences.” Please give more details. This sentence is not clear.

Response: Thank you for your comments and suggestions. We have added more information about aggression to the manuscript, please find the detailed information in the revised manuscript.

Comment 9: - “CEM and aggression are still unclear”; Why? Please give more details. Have you carried out a review? (more than 206000 studies); please see https://scholar.google.com/scholar?hl=pt-PT&as_sdt=0%2C5&q=Childhood+emotional+maltreatment&btnG=

Response: Thank you for your comments and suggestions. We have rewritten this sentence  in the manuscript, such as Some studies have explored the relationships between physical/sexual abuse and aggression (Chen & Qin, 2020; Guo & Chen, 2015) and the relationships between CEM and aggression have also been confirmed by some studies (e.g., Milletich et al., 2010), while the pathways from CEM to aggression remain unclear, particularly in the Chinese cultural context. .

Comment 10: -Please present study objectives, research questions and study hypothesis all together at the end of introduction.

Response: Thank you for your comments and suggestions. We have revised the section of in current study, please find the detailed information in the revised manuscript.

Comment 11: -Lines 99-102: please avoid this type of sentences since authors have not carried out a systematic review. Please rephrase: “Although the relationships between child maltreatment, aggression, resilience, and self-esteem have been explored by some studies, the roles of resilience and self-esteem in the relationships between child maltreatment and aggression are unclear, particularly CEM.” Unclear? Why? References? Please not use this type of sentences. References are missing.

Response: Thank you for your comments and suggestions. We have rewritten these sentences  in the manuscript, please find the detailed information in the revised manuscript. Such as Although the relationships between child maltreatment, aggression, resilience, and self-esteem have been explored by some studies, the roles of resilience and self-esteem in the relationships between child maltreatment and aggression are needed further exploration, particularly in the relationships between CEM and aggression. .

Comment 12: -Please not use this sentence: “few studies have explored the parallel mediating effects of resilience and self-esteem in the relationships between CEM and aggression” Few studies? How do you know? Please not use this type of sentences without developing a strong argumentation. References are missing.

Response: Thank you for your comments and suggestions. We have revised these sentences in the manuscript, please find the detailed information in the revised manuscript. For example, Moreover, resilience and self-esteem mediated the relationships between child maltreatment and later development, respectively (Guo et al., 2015; Thoma et al., 2021), which suggests that resilience and self-esteem may be potential mediators in adverse childhood experiences and later maladaptation (Dambacher et al., 2021; Yang et al., 2021)..

Comment 13: -“Although the relationships between CEM and aggression has been explored by some studies, the pathways from CEM and aggression are still unclear.” Please rewrite. References are also missing.

Response: Thank you for your comments and suggestions. We have revised this sentence in the manuscript, please find the detailed information in the revised manuscript.

Comment 14: -Please cite some systematic reviews on the present topic (or related topics) in introduction. The new references should also be cited in discussion.

Response: Thank you for your comments and suggestions. We have revised this issue in the manuscript, please find the detailed information in the revised manuscript.

Methods

Comment 15: -Please create a separated section for Ethical approval

Response: Thank you for your comments and suggestions. We have created a separated section for Ethical approval in the manuscript, please find the detailed information in the revised manuscript.

-3.1. Participants and procedure

Comment 16: - “The current study recruited 900 participants from three universities in Liaoning Province of China”; When?

Response: Thank you for your comments and suggestions. We have revised these issues in the manuscript and presented a separate section of procedure in the method part, such as “There were several steps for collecting data in the current studies. First, the first author presented the aims and processes of the current study to four potential cooperators, and three of them agreed to collect data in their universities. Second, the authors, respectively, went to these three universities and sent questionnaires to 900 participants initially. All participants were acknowledged about the research before data collection process, and informed consent attached at the beginning of the measure booklets, which to inform them about the purposes of the study and ensure them that their answers would only be used anonymously for research purposes. Third, all participants completed the questionnaires in classrooms within 20 minutes, and all of them received a small gift worth 5RMB ($0.74).”.

Comment 17: -Please present the administered tools in a supplementary file, such as Childhood Trauma Questionnaire–Short Form, the CTQ-SF or the Aggression Questionnaire, etc.

Response: Thank you for your comments and suggestions. We have added the tools to a supplementary file and submitted in the website.

Comment 18: -Methods are incomplete, please present how each one of these tools were administered. Self-administration? Where? When? How? Who?

Response: Thank you for your comments and suggestions. We have revised these issues in the manuscript and presented a separate section of procedure in the method part, such as “There were several steps for collecting data in the current studies. First, the first author presented the aims and processes of the current study to four potential cooperators, and three of them agreed to collect data in their universities. Second, the authors, respectively, went to these three universities and sent questionnaires to 900 participants initially. All participants were acknowledged about the research before data collection process, and informed consent attached at the beginning of the measure booklets, which to inform them about the purposes of the study and ensure them that their answers would only be used anonymously for research purposes. Third, all participants completed the questionnaires in classrooms within 20 minutes, and all of them received a small gift worth 5RMB ($0.74).”.

Comment 19: -Please consider the presentation of point 3.2. Measures in a Tabular format.

Response: Thank you for your comments and suggestions. We have revised this issue in the manuscript, please find the detailed information in the revised manuscript.

Comment 20: -3.3. Data analysis: SPSS 22 and AMOS 22 – please correctly cite SPSS and AMOS. For instance see, https://www.ibm.com/support/pages/how-cite-ibm-spss-statistics-or-earlier-versions-spss

Response: Thank you for your comments and suggestions. We have revised this issue in the manuscript, please find the detailed information in the revised manuscript.

Comment 21: -How were the models validated?

Response: Thank you for your comments and suggestions. We used multiple model fit indexes to validate the models, such as “multiple model fit indexes were used to compare models in the current study, including root-mean-square error of approximation (RMSEA, 0.080 or less), standardized root-mean-square-residual (SRMR,0.080 or less), comparative fit index (CFI, 0.900 or more), normative fit index (NFI, 0.900, or more), goodness fit index (GFI, 0.900 or more), tucker-lewis index (TLI, 0.900 or more) (Hu et al., 1999).”

  1. Results

Comment 22: - Please check the presentation of all statistical findings in APA guidelines.

Response: Thank you for your comments and suggestions. We have checked the presentation of all statistical findings and revised this issue in the manuscript, please find the detailed information in the revised manuscript.

Comment 23: - Please check the format of figures and Tables in instructions to authors.

Response: Thank you for your comments and suggestions. We have checked the format of figures and tables and revised this issue in the manuscript, please find the detailed information in the revised manuscript.

  1. Discussion

Comment 24: -Please cite some new systematic reviews on the present topic

Response: Thank you for your comments and suggestions. We have cited more systematic review in the Discussion part. Please find the detailed information in the revised manuscript.

Comment 25: -Please cite more related or similar studies in discussion

Response: Thank you for your comments and suggestions. We have cited more related or similar studies in the Discussion part. Please find the detailed information in the revised manuscript.

Comment 26: -Please clearly reply to study objectives, research questions and study hypothesis in discussion.

Response: Thank you for your comments and suggestions. We have revised this issue in the manuscript. Please find the detailed information in the revised manuscript.

Comment 27: -Please create a section about study strengths at the end of discussion; please explain in this section. What this study adds in relation to the state of art?

Response: Thank you for your comments and suggestions. We have added more information about study strengths at the end of discussion to the manuscript, please find the detailed information in the revised manuscript.

Conclusion

Comment 28: -Conclusion should reply to study objectives.

Response: Thank you for your comments and suggestions. We have revised this issue in the manuscript. Please find the detailed information in the revised manuscript.

Comment 29: - “Findings underscore the importance of enhancing resilience and self-esteem in interventions designed to reduce aggression of college students who were emotion-ally maltreated in childhood.” Please create a section about practical implications at the end of discussion.

Response: Thank you for your comments and suggestions. We have added a section about practical implication at the end of discussion to the manuscript. Please find the detailed information in the revised manuscript.

References

Comment 30: -Please check the format of references in instructions to authors.

Response: Thank you for your comments and suggestions. We have checked the format of references in the manuscript.

Reviewer 3 Report

Thanks for inviting me to review this paper that examined mediation relationships among childhood emotional maltreatment, resilience, self-esteem, and aggression. Although this line of research has been well replicated, the present paper provided some novel empirical results regarding the mediation effects of resilience and self-esteem. I have some comments when reading this paper. 

1. Please refer to a recent paper published by Morgan CA et al. (Children 2022;9:27), where there is some discussion on the orientations of measured resilience (i.e., trait, outcome, or process). It may be helpful to better situate the present study if the authors can comment on the operational definition of resilience in the analysis.

2. Concerning the analytic diagram, I wonder whether there should be serial mediating effects of self-esteem and resilience sequentially intervening in the link between childhood emotional maltreatment and aggression. A recent paper on a group of Chinese college students (Anal Soc Issues Public Policy. 2022;10.1111/asap.12314. doi:10.1111/asap.12314) has provided evidence on the serial relationship between self-esteem and resilience. If so, the authors should test serial mediating effects when incorporate these two mediators in the analysis.  

Author Response

Dear Editors and Reviewers,

Thank you for your letter and for the reviewer’s comments concerning our manuscript entitled “Resilience and self-esteem mediated associations between childhood emotional maltreatment and aggression” (Manuscript ID: behavsci-1858696) in Behavioral Sciences. Those comments are all valuable and very helpful for revising and improving our manuscript, as well as the important guiding significance to our future studies. We have studied the comments carefully and have made corrections which we hope meet with approval.

The main corrections in the manuscript and the responses to the reviewers’ comment are as follows.

Responses to the reviewers’ comments:

Reviwer#3: Comments and Suggestions for Authors

Comment 0: Thanks for inviting me to review this paper that examined mediation relationships among childhood emotional maltreatment, resilience, self-esteem, and aggression. Although this line of research has been well replicated, the present paper provided some novel empirical results regarding the mediation effects of resilience and self-esteem. I have some comments when reading this paper.

Comment 1: Please refer to a recent paper published by Morgan CA et al. (Children 2022;9:27), where there is some discussion on the orientations of measured resilience (i.e., trait, outcome, or process). It may be helpful to better situate the present study if the authors can comment on the operational definition of resilience in the analysis.

Response: Thank you for your comments and suggestions. We have read this study, and cited it in the manuscript. Please find the detailed information in the revised manuscript.

Comment 2: Concerning the analytic diagram, I wonder whether there should be serial mediating effects of self-esteem and resilience sequentially intervening in the link between childhood emotional maltreatment and aggression. A recent paper on a group of Chinese college students (Anal Soc Issues Public Policy. 2022;10.1111/asap.12314. doi:10.1111/asap.12314) has provided evidence on the serial relationship between self-esteem and resilience. If so, the authors should test serial mediating effects when incorporate these two mediators in the analysis.

Response: Thank you for your comments and suggestions. Self-esteem and resilience may serially mediate the relationships between CEM and aggression, while the current study is a cross-sectional study which cannot explore the directions between self-esteem and resilience. As a result, the current study did not perform the serially mediation roles of self-esteem and resilience in the relationships between CEM and aggression. Future longitudinal studies may explore the serially mediation roles of self-esteem and resilience in the relationships between CEM and aggression.

Reviewer 4 Report

The title does not correctly describe the study.

There is a deficit in updating the theoretical framework of the introduction (at least 40% of the last 4 years).

Important methodological deficiencies:

- The sample is not justified (university students).

- Lack of information on the procedure (guidelines for the uniform application of the tests, norms, rejection rate, etc.).

The planted model contributes practically nothing new.

The conclusions are known (nothing new).

Author Response

Dear Editors and Reviewers,

Thank you for your letter and for the reviewer’s comments concerning our manuscript entitled “Resilience and self-esteem mediated associations between childhood emotional maltreatment and aggression” (Manuscript ID: behavsci-1858696) in Behavioral Sciences. Those comments are all valuable and very helpful for revising and improving our manuscript, as well as the important guiding significance to our future studies. We have studied the comments carefully and have made corrections which we hope meet with approval.

The main corrections in the manuscript and the responses to the reviewers’ comment are as follows.

Responses to the reviewers’ comments:

Reviewer#4: Comments and Suggestions for Authors

Comment 1: The title does not correctly describe the study.

Response: Thank you for your comments and suggestions. We have revised this issue in the manuscript. Please find the detailed information in the revised manuscript.

Comment 2: There is a deficit in updating the theoretical framework of the introduction (at least 40% of the last 4 years).

Response: Thank you for your comments and suggestions. We have added more new studies to the manuscript and revised this issue in the manuscript. Please find the detailed information in the revised manuscript.

Important methodological deficiencies:

Comment 3: - The sample is not justified (university students).

Response: Thank you for your comments and suggestions. We have added more information about this issue to the manuscript, such as “Additionally, college period is one of the most important period in the whole life, and experiences in childhood may have more effects on individuals’ development in this period, similarly, college students may be an important component for social development, particularly in developing countries, which suggests that we should pay attention to college students’ individual development, such as aggression. The current study, therefore, attempts to explore the relationships between CEM and aggression in the Chinese cultural context.”.

Comment 4: - Lack of information on the procedure (guidelines for the uniform application of the tests, norms, rejection rate, etc.).

Response: Thank you for your comments and suggestions. We have revised these issues in the manuscript and presented a separate section of procedure in the method part, such as “There were several steps for collecting data in the current studies. First, the first author presented the aims and processes of the current study to four potential cooperators, and three of them agreed to collect data in their universities. Second, the authors, respectively, went to these three universities and sent questionnaires to 900 participants initially. All participants were acknowledged about the research before data collection process, and informed consent attached at the beginning of the measure booklets, which to inform them about the purposes of the study and ensure them that their answers would only be used anonymously for research purposes. Third, all participants completed the questionnaires in classrooms within 20 minutes, and all of them received a small gift worth 5RMB ($0.74).”.

Comment 5: The planted model contributes practically nothing new.

Response: Thank you for your comments and suggestions. We have added some implications to the manuscript, such as “First, the results showed that CEM was positively associated with aggression in college students, which suggests that CEM may have a long-term effect on individuals’ later development. The governments and communities should provide supports of positively parenting behaviors for Chinese parents and decrease the prevalence of CEM in Chinese samples. Second, the results showed that resilience and self-esteem parallel mediated the relationships be-tween CEM and aggression, which suggests that resilience and self-esteem may be possible mediators for pathways from CEM to aggression. The schools should provide strategies for increasing levels of resilience and self-esteem in survivors of CEM. For example, schools should provide courses about communication skills, which may increase the levels of communication skills in survivors of CEM, and in turn, contribute to high levels of resilience. Additionally, teachers may give more positive feedbacks to the survivors of CEM and help them rebuild their evaluations of self, which may increase the levels of self-esteem in survivors of CEM.”.

Comment 6: The conclusions are known (nothing new).

Response: Thank you for your comments and suggestions. We have added some strengths of the current study to the manuscript, such as “First, the current study verifies the relationships between CEM and aggression in a sample of Chinese college students, which may broaden the scopes of childhood maltreatment studies across nations and cultures. Second, the cur-rent study examines the roles of resilience and self-esteem in the relationships between CEM and aggression, which may provide possible mechanisms linking childhood maltreatment and later development. Third, the current study may provide possible strategies for interventions of CEM in college students.”.

Round 2

Reviewer 2 Report

Congratulations. The paper has been improved.

Some minor recommendations:

-          After carrying out all corrections (please see below), please proofread the paper once more.

Abstract

Please clarify “physical and sexual” or “physical or sexual” or “physical and/or sexual” (instead of “physical/sexual”. Please check all the paper for “physical/sexual”.

The title of Line 96 should appear in the next page.

Please cite the supplementary files/Annexes or Appendices in the manuscript between brackets (e.g., supplementary file 1, supplementary file 2, etc.). Please check how to cite supplementary files or annexes in the instructions for authors. For instance, The “Childhood Trauma Questionnaire–Short Form (supplementary file …) (CTQ-SF; Bernstein 174 & Fink, 1998; Bernstein et al., 2003).” Please present the original version of all administered tools (English translation) in supplementary files/Annexes or Appendices, which must be cited in materials and methods.

Line 279: please move the title for the next page.

Table 3: For instance, please use “0.978” instead of “.978” (i.e., please use “0.” Instead of “.”). Please use the same format for decimal place in all the manuscript. Please check instruction for authors. Please check the formats of number and decimal places in all the paper.

Figures and Tables: Please present the full meaning of all abbreviations below figures and Tables. For instance, Figure 1: RS, EA, EN, CEM, AG, PA, OA, AN, HO.

Discussion

-          Please consider the presentation of limitations, practical implications and study strengths in separated subheadings, because it will increase text readability and intelligibility.

Author Response

Dear Editors and Reviewers,

Thank you for your letter and for the reviewer’s comments concerning our manuscript entitled “Resilience and self-esteem mediated associations between childhood emotional maltreatment and aggression” (Manuscript ID: behavsci-1858696) in Behavioral Sciences. Those comments are all valuable and very helpful for revising and improving our manuscript, as well as the important guiding significance to our future studies. We have studied the comments carefully and have made corrections which we hope meet with approval.

The main corrections in the manuscript and the responses to the reviewers’ comment are as follows.

Responses to the reviewers’ comments:

Reviewer#2: Comments and Suggestions for Authors

Comment 1: Congratulations. The paper has been improved.

Response: Thank you very much. We so appreciate your comments and suggestions that help us improve the manuscript.

Some minor recommendations: 

Comment 2: After carrying out all corrections (please see below), please proofread the paper once more.

Abstract

Comment 3: Please clarify “physical and sexual” or “physical or sexual” or “physical and/or sexual” (instead of “physical/sexual”. Please check all the paper for “physical/sexual”. 

Response: Thank you very much for your suggestion. We have revised this issue throughout the manuscript.

Comment 4: The title of Line 96 should appear in the next page.

Response: Thank you very much for your suggestion. We have revised this issue in the revised manuscript.

Comment 5: Please cite the supplementary files/Annexes or Appendices in the manuscript between brackets (e.g., supplementary file 1, supplementary file 2, etc.). Please check how to cite supplementary files or annexes in the instructions for authors. For instance, The “Childhood Trauma Questionnaire–Short Form (supplementary file …) (CTQ-SF; Bernstein 174 & Fink, 1998; Bernstein et al., 2003).” Please present the original version of all administered tools (English translation) in supplementary files/Annexes or Appendices, which must be cited in materials and methods.

Response: Thank you very much for your suggestion. We have added these information to the revised manuscript. Please find the detailed information in the revised manuscript.

Comment 6: Line 279: please move the title for the next page.

Response: Thank you very much for your suggestion. We have revised this issue in the revised manuscript.

Comment 7: Table 3: For instance, please use “0.978” instead of “.978” (i.e., please use “0.” Instead of “.”). Please use the same format for decimal place in all the manuscript. Please check instruction for authors. Please check the formats of number and decimal places in all the paper.

Response: Thank you very much for your suggestion. We have revised this issue throughout the manuscript. Please find the detailed information in the revised manuscript.

Comment 8: Figures and Tables: Please present the full meaning of all abbreviations below figures and Tables. For instance, Figure 1: RS, EA, EN, CEM, AG, PA, OA, AN, HO. 

Response: Thank you very much for your suggestion. We have added these information to the revised manuscript. Please find the detailed information in the revised manuscript.

Discussion

Comment 9: Please consider the presentation of limitations, practical implications and study strengths in separated subheadings, because it will increase text readability and intelligibility.

Response: Thank you very much for your suggestion. We have revised this issue in the Discussion part. Please find the detailed information in the revised manuscript.

Reviewer 3 Report

I have no more questions.

Author Response

Dear Editors and Reviewers,

Thank you for your letter and for the reviewer’s comments concerning our manuscript entitled “Resilience and self-esteem mediated associations between childhood emotional maltreatment and aggression” (Manuscript ID: behavsci-1858696) in Behavioral Sciences. Those comments are all valuable and very helpful for revising and improving our manuscript, as well as the important guiding significance to our future studies. We have studied the comments carefully and have made corrections which we hope meet with approval.

The main corrections in the manuscript and the responses to the reviewers’ comment are as follows.

Responses to the reviewers’ comments:

Reviwer#3: Comments and Suggestions for Authors

Comment 0: I have no more question.

Response: Thank you very much. We so appreciate your comments and suggestions that help us improve the manuscript.

Reviewer 4 Report

The work has improved substantially following the changes, but:

- There are too many keywords.

- 40% of the references of the last 4 years are still missing (has been updated, but not sufficiently).

- The justification of the sample is not scientifically contrasted. In addition, there is a sample of NON-university adolescents who are excluded from the study without justification (and perhaps this is more in line with the objective).

- The selection of the sample is not justified by any specific criteria.  What the authors indicate is not a scientific criterion.

- There is still a lack of procedural information.

- It is reiterated that the study does not contribute anything new, as the importance of resilience and self-esteem for CEM and aggression is well known.

- All identifying details of the ethics committee approval should be provided.

Author Response

Dear Editors and Reviewers,

Thank you for your letter and for the reviewer’s comments concerning our manuscript entitled “Resilience and self-esteem mediated associations between childhood emotional maltreatment and aggression” (Manuscript ID: behavsci-1858696) in Behavioral Sciences. Those comments are all valuable and very helpful for revising and improving our manuscript, as well as the important guiding significance to our future studies. We have studied the comments carefully and have made corrections which we hope meet with approval.

The main corrections in the manuscript and the responses to the reviewers’ comment are as follows.

Responses to the reviewers’ comments:

Reviewer#4: Comments and Suggestions for Authors

The work has improved substantially following the changes, but:

- Comment 1: There are too many keywords.

Response: Thank you very much for your suggestion. We have reduced the number of keywords, please find the detailed information in the revised manuscript, such as childhood maltreatment; later development; externalizing problems; resilience; Chinese cultural context.

- Comment 2: 40% of the references of the last 4 years are still missing (has been updated, but not sufficiently).

Response: Thank you very much for your comment. We have added much more references of the last 4 years to the manuscript, please find the detailed information in the revised manuscript. The rate of references of the last 4 years (2018-2022) is 45.31%(29/64).

- Comment 3: The justification of the sample is not scientifically contrasted. In addition, there is a sample of NON-university adolescents who are excluded from the study without justification (and perhaps this is more in line with the objective).

Response: Thank you very much for your comment and suggestion. We have added this information to limitation part of the manuscript, please find the detailed information in the revised manuscript, such as Fifthly, individuals who have experienced CEM may have poor academic achievement, which may contribute to their low levels of educational years. The current study explored the relationships between CEM, aggression, self-esteem, and resilience in a sample of Chinese college students, which may not provide the whole picture of those relationships. Future studies should recruit non-university adolescents or young adults which may help us better under-standing the pathways from CEM and aggression..

- Comment 4: The selection of the sample is not justified by any specific criteria.  What the authors indicate is not a scientific criterion.

Response: Thank you very much for your comment and suggestion. We have added this information to limitation part of the manuscript, please find the detailed information in the revised manuscript, such as Secondly, the current study used convenience sampling techniques to recruit the participants which may not be a representative sample. Future studies should recruit participants based on random sampling or stratified sample techniques..

- Comment 5: There is still a lack of procedural information.

Response: Thank you very much for your comment. We have added more information about procedures to the manuscript. Please find the detailed information in the revised manuscript.

- Comment 6: It is reiterated that the study does not contribute anything new, as the importance of resilience and self-esteem for CEM and aggression is well known.

Response: Thank you very much for your comment. Although we have well known about the importance of resilience and self-esteem for CEM and aggression, the current study may replicate these results or relationship across samples and cultures, which may provide more evidence for relationships among these four variables.

- Comment 7: All identifying details of the ethics committee approval should be provided.

Response: Thank you very much for your comment. We have added more information about procedures to the manuscript. Please find the detailed information in the revised manuscript.

Round 3

Reviewer 4 Report

It is not understood why changes from the first review were not made.

Ethics committee information is incomplete.

Most of the comments have been put forward as limitations of the study, with the work having many limitations.